# Hormone Regulation in Testicular Development and Function

**DOI:** 10.3390/ijms25115805

**Published:** 2024-05-26

**Authors:** Lu Li, Wanqing Lin, Zhaoyang Wang, Rufei Huang, Huan Xia, Ziyi Li, Jingxian Deng, Tao Ye, Yadong Huang, Yan Yang

**Affiliations:** 1Department of Cell Biology, Jinan University, Guangzhou 510632, China; lilu2022@stu2022.jnu.edu.cn (L.L.); lwq2022@stu2022.jnu.edu.cn (W.L.); wzy1003@stu2021.jnu.edu.cn (Z.W.); sophie12@stu2022.jnu.edu.cn (R.H.); xiahuan@stu2019.jnu.edu.cn (H.X.); lzy2023@stu2023.jnu.edu.cn (Z.L.); dengjingxian@stu2021.jnu.edu.cn (J.D.); taoyhust@stu2022.jnu.edu.cn (T.Y.); 2Guangdong Province Key Laboratory of Bioengineering Medicine, Guangzhou 510632, China; 3National Engineering Research Center of Genetic Medicine, Guangzhou 510632, China

**Keywords:** hormones, spermatogenesis, testis, HPG axis, male reproduction

## Abstract

The testes serve as the primary source of androgens and the site of spermatogenesis, with their development and function governed by hormonal actions via endocrine and paracrine pathways. Male fertility hinges on the availability of testosterone, a cornerstone of spermatogenesis, while follicle-stimulating hormone (FSH) signaling is indispensable for the proliferation, differentiation, and proper functioning of Sertoli and germ cells. This review covers the research on how androgens, FSH, and other hormones support processes crucial for male fertility in the testis and reproductive tract. These hormones are regulated by the hypothalamic–pituitary–gonad (HPG) axis, which is either quiescent or activated at different stages of the life course, and the regulation of the axis is crucial for the development and normal function of the male reproductive system. Hormonal imbalances, whether due to genetic predispositions or environmental influences, leading to hypogonadism or hypergonadism, can precipitate reproductive disorders. Investigating the regulatory network and molecular mechanisms involved in testicular development and spermatogenesis is instrumental in developing new therapeutic methods, drugs, and male hormonal contraceptives.

## 1. Introduction

The decline in male sperm count has been a common phenomenon worldwide. From 1973 to 2018, the mean sperm concentration and total sperm count among unselected men from all continents decreased by 51.6% and 62.3%, respectively [1,2]. This large and sustained decline is now considered a major public health issue, and the relationship between sperm count and infertility has received widespread attention. Infertility or reduced fertility can be attributed to endocrine diseases, testicular dysfunction, and poor lifestyle factors such as unhealthy diet and alcohol consumption. The endocrine system serves as the primary regulator of reproductive function, and a delicate hormonal balance and crosstalk are critical for testicular development and spermatogenesis.

The testes, being vital male reproductive organs, are responsible for the production of sperm and male sex hormones. Hormones play a vital role in regulating testicular development and function in males from fetal life through adulthood. Proper hormone regulation is essential for maintaining male reproductive health, which includes sexual maturation, germ cell production, and steroidogenesis. Imbalances in hormonal regulation associated with testicular disorders can result in a variety of health problems, such as infertility, sexual dysfunction, and testicular cancer. Therefore, having a thorough understanding of the hormonal regulation of testicular development and function is crucial for the accurate diagnosis and treatment of these disorders. This review aims to provide an overview of the hormones involved in the regulation of male reproduction, as well as recent advancements in related clinical and applied research.

## 2. Testis Physiology

The testes comprise multiple compartments, including the seminiferous tubules, interstitial cells (Leydig cells), and supporting cells (Sertoli cells). Testis physiology encompasses the intricate processes involved in sperm production, testosterone secretion, and hormonal regulation. It involves the coordinated functioning of various cells and hormones to maintain male reproductive health and fertility (Figure 1). Aside from sperm production and testosterone production, the testes also play a role in the regulation of sexual function. The testes produce other hormones, such as inhibin, which helps regulate follicle-stimulating hormone (FSH) levels, and estrogen, albeit in small amounts. Understanding testis physiology is crucial for diagnosing and treating male reproductive disorders and infertility. Any disruption in hormone production or spermatogenesis can lead to fertility issues, hormonal imbalances, and sexual dysfunctions.

### 2.1. Testicular Structure

#### 2.1.1. Seminiferous Tubule

Seminiferous tubules are home to germ cells, which are crucial for spermatogenesis, as well as Sertoli cells and peritubular myoid cells (PMCs). Sertoli cells offer support and nutrition to developing sperm, and PMCs surround the tubule’s external wall. The most optimal energy source for developing germ cells, the lactate molecule, is provided by Sertoli cells, which also furnish necessary growth factors and chemokines. The microenvironment created by neighboring Sertoli cells is vital for maintaining germ cell growth and initiating differentiation [3,4]. More specifically, Sertoli cells are indispensable for managing androgen production within the testis and encouraging the secretion of a range of bioactive peptides [5]. These peptides facilitate communication between Sertoli cells and germ cells to support spermatogenesis and fertility [4,5]. PMCs play a role in regulating sperm and luminal fluid transport and secrete growth factors and an extracellular matrix to uphold the niche of spermatogonial stem cells (SSCs). For example, the glial cell-derived neurotrophic factor (GDNF) produced by PMCs is instrumental in the development of undifferentiated spermatogonia cells in vivo [6].

#### 2.1.2. Interstitial Tissue

Interstitial tissue contains Leydig cells, which are responsible for testosterone production, as well as blood vessels, nerves, and connective tissue. The mammalian Leydig cell is the primary site for testosterone secretion, and this hormone is crucial for spermatogenesis. Additionally, two types of macrophages, peritubular macrophages, and interstitial macrophages, also contribute to testicular function [7]. These macrophages differ in their location and function, but overall, they play a role in inducing spermatogonial proliferation and differentiation [8], providing a suitable niche for Leydig cells [9], and maintaining the immune privileges of the testis with the blood–testis barrier (BTB) [10].

#### 2.1.3. Blood–Testis Barrier (BTB)

The BTB is a unique structure consisting the gap junctions, tight junctions (TJs), desmosomes, and ectoplasmic specializations between Sertoli cells [11] (Figure 1). The tight junction proteins composing the BTB include the claudins (CLDNs), junctional adhesion molecules (JAMs), etc., and they bind to actin through the zonula occluden-1 (ZO-1), ZO-2, and ZO-3 [12,13]. The BTB is one of the strictest tissue barriers found in mammals and divides the seminiferous epithelium into the basal part and an adluminal part. This intricate barrier segregates the seminiferous tubules from the bloodstream, effectively restricting the diffusion of water, ions, electrolytes, paracrine factors, hormones, and other exogenous biomolecules through both the paracellular and transcellular pathways. Consequently, it creates a specialized microenvironment that is immune-privileged and conducive to germ cell development [14]. The BTB is crucial for successful spermatogenesis as it helps safeguard developing sperm from autoimmune reactions. However, the presence of this barrier also poses a challenge for the development of drugs that target the testicles.

### 2.2. Testicular Function

#### 2.2.1. Spermatogenesis

Spermatogenesis is a complex process that involves the development of male germ cells, known as spermatogonia, into fully mature spermatozoa, or sperm. This process occurs within the seminiferous tubules and consists of three main stages: the mitosis of spermatogonia, followed by the meiosis of spermatocytes, and finally the morphological transformation of spermatids into spermatozoa. During spermatogenesis, the germ cells undergo movement within the seminiferous epithelium, leading to the restructuring of the junctions between Sertoli cells and germ cells, as well as between Sertoli cells themselves at the BTB [11]. After spermatogenesis is completed, mature sperm will be released from the basal part of the seminiferous epithelium into the seminiferous tubule lumen. Additionally, the premature shedding of germ cells from Sertoli cells indicated a failure of spermatogenesis [15]. Sertoli cells play a critical role in spermatogenesis, as they provide a specialized environment for sperm cells. These cells contribute to the maintenance of the BTB and facilitate the proper development and release of mature spermatozoa. Understanding the intricate mechanisms involved in spermatogenesis is essential for elucidating the causes of male infertility and developing potential therapeutic interventions.

#### 2.2.2. Testosterone Production

Testosterone is the primary androgen in the male body, and its levels directly affect spermatogenesis [16]. Testosterone is primarily synthesized and secreted by Leydig cells in the testes through a series of enzymatic reactions. The regulation of testosterone levels is governed by the negative feedback mechanisms of the HPG axis, as well as local factors within the testes. The proper development of Leydig cells during puberty is essential for initiating spermatogenesis and promoting secondary sexual characteristics in males [17]. Studies have shown that the transplantation of stem Leydig cells (SLCs) in Leydig cell-disrupted or aging models can help restore testosterone production, thereby accelerating meiosis and germ cell recovery [18,19].

In summary, testicular physiology comprises an intricate interrelationship involving various cell types, hormones, and signaling pathways, all of which play a crucial role in regulating spermatogenesis and hormone synthesis. These processes are fundamental for maintaining male fertility and overall reproductive health.

## 3. Hormone Regulation in Testicular Development and Function

The hypothalamic–pituitary–gonadal (HPG) axis is a vital system involved in hormonal regulation. It consists of the hypothalamus, which releases gonadotropin-releasing hormone (GnRH); the pituitary gland, which secretes LH and FSH; and the gonads (testes in males and ovaries in females). In males, GnRH from the hypothalamus stimulates the pituitary gland to release LH and FSH. LH acts on Leydig cells in the testes, stimulating the production of testosterone and INSL3. Testosterone plays a crucial role in the development of male secondary sexual characteristics, libido, and spermatogenesis. Insulin-like peptide 3 (INSL3) is involved in the descent of the testes during fetal development.

FSH acts on Sertoli cells in the testes, supporting spermatogenesis. Sertoli cells secrete inhibin, which inhibits the release of FSH from the pituitary gland, helping to regulate the levels of FSH (Figure 2). Sertoli cells also secrete anti-Müllerian hormone (AMH), which plays a role in the regression of female reproductive structures during male development. Imbalances in these hormones may lead to decreased fertility, infertility, or other reproductive/non-reproductive system diseases (Table 1).

### 3.1. Testosterone

Testicular androgens, specifically testosterone and its metabolite, play a crucial role in male fertility and are essential for normal masculinization, testis function, and other androgenic targets such as muscle, fat, and bone. As with all steroid hormones, the initial precursor required for the production of androgen is cholesterol. The established pathway involves a series of enzymatic reactions that convert the steroid precursors into testosterone and dihydrotestosterone (DHT). A critical enzyme in this process is 17β-hydroxysteroid dehydrogenase type 3 (HSD17B3).

Testosterone exerts its effects by binding to the androgen receptor (AR). In the classical signaling pathway, AR in the cytoplasm undergoes conformational changes to androgen. Subsequently, it dissociates from chaperone proteins like heat shock proteins (HSPs), and the receptor–ligand complex enters the nucleus. Within the nucleus, the complex recruits coactivators or corepressors to bind with androgen response elements (AREs) in the promoter regions of targeted genes, thereby activating or inhibiting gene transcription [26] (Figure 3). On the other hand, the non-classical testosterone signaling pathway becomes active when intratesticular testosterone levels are low, typically below 250 nM, and testosterone rapidly binds to the membrane AR and triggers a cascade reaction [27]. This pathway predominantly functions by promoting the phosphorylation and activation of ERK1/2 and CREB [27]. During this process, AR interacts with and activates Src kinase at the plasma membrane [28]. Src activation leads to phosphorylation and activation of EGFR, which, in turn, activates MAPK cascades (Ras-Raf-MEK-ERK), resulting in the activation of transcription factors such as cAMP-response element-binding (CREB) proteins and a series of downstream effects [28,29] (Figure 3). Androgens regulate the expression of a series of genes through these two pathways, directly affecting the function of Sertoli cells or acting on germ cells via paracrine mechanisms to regulate sperm production.

For Sertoli cells, AR signaling plays a role in regulating Sertoli cells to the cessation of proliferation and the initiation of differentiation. This is mainly through regulating Smad2/3 signaling [30], down-regulating the expression of AMH [31,32], and promoting the formation of cytoskeleton [33,34]. There is controversy over the effect of testosterone on Sertoli cell proliferation, and this effect appears to be species-specific and stage-specific [35,36,37]. Moreover, inhibiting the action of testicular hormones or mutations in the androgen receptor can lead to the increased permeability of the BTB [38,39,40]. Testosterone regulates the expression of TJ proteins (including the membrane-associated guanylate kinase (MAGUK) family, Claudins, and Occludins) [41,42,43,44], tissue-type plasminogen activator (tPA) (which is involved in BTB degradation) [45,46], basal ectoplasmic specialization (ES) proteins [47], and gap junction proteins (including Connexin 43) directly or indirectly to maintain the integrity of the BTB and support its renewal [48].

Also, the process of differentiation from SSCs to the eventual release of sperm is guaranteed by AR signaling. The effect of testosterone on male reproduction is also reflected in germ cells. After the differentiation of gonocytes into SSCs, they are influenced by the signaling network that triggers self-renewal and differentiation. Testosterone regulates the self-renewal and differentiation of SSC by modulating the expression of multiple genes, including Wnt5a [49], promyelocytic leukemia zinc finger (Plzf) [50], insulin-like growth factor 3 (Igf3) [51], and reproductive homeobox 5 (Rhox5) [52]. What is more, testosterone plays a key role in spermatogenesis by participating in meiosis. The absence of testosterone stimulation leads to the failure of the transformation of round sperm cells in stages VII and VIII of the rat spermatogenic cycle during spermatogenesis [53,54]. On the one hand, the lack of AR signals leads to the dysfunction of the chromosome, mainly manifested in the tissues with double-strand breaks (DSBs) repair and chromosome synapsis [37]. After androgen deprivation or the knockdown of AR in Sertoli cells, the expression and function of proteins essential for DSB repair, synaptonemal complex formation, spindle dynamics, and dynactin complex assembly are compromised, ultimately disrupting the process of meiosis [55,56]. On the other hand, testosterone prevents the apoptosis of germ cells during mitosis. Impaired AR signal transduction in Sertoli cells is caused by oxidative stress due to chromosomal dysfunction or breakage, resulting in cell apoptosis [57]. Furthermore, reduced testosterone levels elevate ubiquitin carboxyl-terminal hydrolase L1 (UCHL1) expression in spermatogonia, thereby promoting the ubiquitination of the apoptosis factor p53 [58]. Testosterone also promotes the phagocytic clearance of apoptotic germ cells by Sertoli cells, possibly by increasing the expression of miR-471-5p and Elmo1 [59,60,61]. The apoptosis of numerous germ cells may trigger the inner meiotic checkpoint, ultimately resulting in the termination of meiosis in the prophase I [37].

Additionally, the role of androgen signaling also includes maintaining the Sertoli cell–spermatid adhesion to prevent spermatid shedding from the epithelium and regulating the release of sperm. Low testosterone levels in rat testicles may lead to the loss of stage VIII and late spermatids [62,63]. Testosterone promotes the attachment of spermatids to Sertoli cells through the non-classical pathway [29]. The activation of Src and FAK is essential to support sperm cell adhesion and sperm release, and both can be phosphorylated during non-classical androgen signaling activation [54,63,64].

Impaired fetal androgen action can lead to disorders of sexual differentiation [65], including the common human congenital disorder testicular dysgenesis syndrome (TDS). TDS is associated with various conditions such as cryptorchidism and hypospadias in male newborns [66,67], impaired spermatogenesis, and testicular germ cell cancer (TGCC) in young adult males [68]. Interestingly, the role of androgens in establishing normal reproductive tract development and the masculinization of anogenital distance (AGD) is limited to a specific developmental window, known as the “masculinization programming window” (MPW). In rats, this window is between E15.5 and E18.5, while in humans, it is postulated to be between 8 and 14 weeks of gestation [69].

### 3.2. FSH

In the past century, it has been acknowledged that FSH plays a vital role in maintaining the normal production of germ cells in males [70]. FSH is a glycoprotein composed of an α-subunit and a β-subunit. The α subunit is shared by TSH, LH, and HCG, while the FSHβ-subunit is unique to FSH [71]. The FSH receptor (FSHR), responsible for binding FSH, is exclusively expressed on the cell membrane of Sertoli cells. FSH-induced signal transduction is mediated by FSHR, and its function reliant on interactions with numerous intracellular effectors.

The action of FSH mainly relies on the cAMP/PKA signaling pathway. FSH promotes Sertoli cell proliferation via the PI3K/Akt pathway and enhances the expression of cell-derived Myc (c-Myc) and type D1 cyclin (cyclin D1) through the cAMP-dependent pathway during fetal development and early postpartum stages [72,73,74]. The suppression of FSH in neonates reduces the number of final Sertoli cells by approximately 40% [75], which in turn affects the quantity of sperm. Furthermore, FSH targets functional factors and transcription factors through the cAMP/PKA pathway to affect Sertoli cell differentiation and apoptosis [76,77]. For example, transcription factors including Krüppel-like factor 4 (KLF4) [78], nuclear factor (NF)-κB [79], and activator protein-1 (AP-1) [80] are involved in the regulation of Sertoli cell differentiation by FSH.

The cAMP/PKA signaling pathway plays a pivotal role in the regulation of FSH on the maintenance of the spermatogonia pool and the differentiation and apoptosis of spermatogonia. FSH stimulates the Sertoli cells to secrete GDNF and fibroblast growth factor 2 (FGF2), which are crucial for the self-renewal of spermatogonia stem cells (SSCs) and the proliferation of undifferentiated spermatogonia, as indicated by studies [81,82,83,84,85]. Furthermore, signaling enhances SSC differentiation through the activation of key factors such as stem cell factor (SCF), steel factor (SLF), bone morphogenetic protein-4 (BMP4), and insulin-like growth factor 3 (IGF3) [51,86,87]. This intricate hormonal interplay ensures the delicate balance between spermatogonia maintenance and differentiation, which is essential for testicular development and function.

FSH is also essential for meiosis. It controls DNA synthesis and meiotic chromosome dynamics through the regulation of activin A, inhibin B, IL-6, and nociception, thus promoting spermatocytes to enter meiosis [88,89,90]. In the early stage of meiosis, FSH protects spermatocytes from apoptosis by inducing the expression of galectin-3 in Sertoli cells and inhibiting the activation of transcription factor AP-1 [91,92]. The knockout of FSHR and FSH β in mice resulted in a decrease in the number of Sertoli cells, spermatogonia, and spermatocytes [93,94].

FSH signaling is intricately linked to testicular development and function, with pathways such as the ERK/MAPK, calcium, and phospholipase A2 pathways identified in FSH-mediated regulation [72,76,95]. Female transgenic mice lacking FSH or its receptor (FSHR) exhibit infertility, while their male counterparts, despite remaining fertile, display reduced testicular size and germ cell count [96,97,98]. Similarly, men with congenital FSH deficiency often face infertility, typically attributed to FSHβ gene mutations [99,100].

In a clinical setting, patients who have undergone hypophysectomy for pituitary tumors may still exhibit autonomous spermatogenesis due to the ligand-independent, constitutive activation of FSHR [101]. This phenomenon underscores the potential for FSH signaling to sustain spermatogenesis independently of pituitary gonadotropins. Oduwole et al.’s research on transgenic mice with a constitutively active FSHR mutant form further supports this notion, as it demonstrated the partial restoration of fertility and the resumption of sperm production despite the absence of LH receptors [102]. These findings suggest that robust FSHR activity can facilitate spermatogenesis without the need for testosterone, challenging the notion that testosterone is an absolute prerequisite for this process [103]. This highlights the pivotal role of FSHR activation in the maintenance of sperm production, offering new insights into the hormonal regulation of testicular function.

### 3.3. Inhibin B

Inhibin B, along with inhibin A, are peptides produced by Sertoli cells within the testes, playing a crucial role in the endocrine feedback regulation of FSH at the pituitary level. Predominant in primates, inhibin B is more abundant in adult serum compared to inhibin A. A significant surge in inhibin B levels is observed postnatally, coinciding with the proliferation of Sertoli cells, followed by a decline until puberty, when FSH stimulation leads to a resurgence [104]. Therefore, inhibin B is more appropriate for the early assessment of testicular function during these developmental stages [105,106]. Additionally, inhibin B is a vital biomarker for spermatogenesis and an indicator of Sertoli cell functionality, closely associated with the size of the Sertoli cell population and testicular volume [107,108]. In cases of oligospermia, there is a strong correlation between inhibin B and FSH levels with sperm concentration, and elevated FSH (>7.8 IU/I) and reduced inhibin B (<92 pg/mL) levels are indicative of a compromise [109,110,111]. This association underscores the importance of inhibin B in the diagnosis and monitoring of male fertility and testicular health.

### 3.4. Activin A

Activin is a dimer glycoprotein belonging to the transforming growth factor β (TGF-β) superfamily. It is composed of two β subunits, which can combine with the α subunit to form inhibin [112]. Unlike inhibin, activin stimulates the pituitary gland to produce FSH. Activin signals are activated through interaction with specific receptors that belong to a serine/threonine kinase family and activate intracellular Smad proteins [113].

Activin A plays multiple roles in the testis. It serves as an important factor in regulating the proliferation of Sertoli cells and gonocytes, which are crucial for the establishment of a balance between Sertoli and germ cells at the beginning of spermatogenesis [114]. The absence of activin A leads to a decrease in the proliferation of fetal Sertoli cells [115], while some gonocytes may bypass the quiescent state and multiply in number [114]. Furthermore, activin A also influences the development and function of the epididymis. Transgenic mice that have been engineered to overexpress the activin antagonist follistatin (FST) in the testis exhibit fluid accumulation and sperm stasis [116]. This result suggested that the blockade of activin signaling in males resulted in impaired testicular duct function, degenerative lesions, and impaired fertility, suggesting that the selective inhibitors of activin signaling could potentially be used for the development of male contraceptives without affecting androgen synthesis and actions. Additionally, the knockout of the activin/inhibin A subunit gene (INHBA) resulted in an abnormal morphology of Wolffian tubes, specifically a failure to develop the characteristic coiling in the epithelium [117]. Recent studies have also revealed that activin A plays a role in regulating androgen biosynthesis in fetal testis by acting on Sertoli cells [118]. Activin A deficiency can also lead to disturbances in the male reproductive system.

### 3.5. Anti-Müllerian Hormone (AMH)

Anti-Müllerian Hormone (AMH) serves as a marker for immature Sertoli cells [119], which are instrumental in the regression of Müllerian ducts in male fetuses, thus preventing the development of the uterus and fallopian tubes [120]. AMH secretion by Sertoli cells remains elevated throughout fetal life and into postnatal development until the onset of puberty [121,122]. As puberty progresses, Sertoli cells transition from a proliferative, immature state to a mature, quiescent one, leading to a marked decrease in AMH levels [123]. This reduction in AMH is linked to the maturation of Sertoli cells and inversely correlates with circulating androgen levels [120].

Serum AMH levels can be utilized as an indicator of testicular presence. A study revealed that serum AMH levels averaged around 48.2 ng/mL in children with normal testes, 11.5 ng/mL in children with abnormal testes, and only 0.7 ng/mL in cases of testicular absence [124]. The AMH assay demonstrates high sensitivity, up to 92%, in detecting the absence of testicular tissue [124]. Consequently, AMH levels are invaluable in differential diagnosis, particularly for conditions such as bilateral cryptorchidism and anorchidism in boys with nonpalpable gonads [119,120,124]. This biomarker provides a reliable tool for assessing testicular function and development, offering insights into the hormonal regulation of testicular health.

### 3.6. Insulin-like Factor 3 (INSL3)

INSL3 belongs to the insulin–relaxin family and is primarily produced in Leydig cells in human males [125]. The expression of INSL3 is directly dependent on the number and differentiation status of Leydig cells and is an ideal biomarker for Leydig cells [126]. During fetal development, INSL3 expression starts after gonadal sex determination and acts by binding to the RelaXin-like Family Peptide receptor 2 (RXFP2), which is expressed by gubernacular ligament mesenchymal cells that connect the testis to the inguinal wall [125]. Their crucial role is reflected in the process of the descent of the fetal testes [125]. During the first phase of testicular descent, INSL3 mediates the outgrowth of the gubernaculum, thereby retaining the fetal testis in the inguinal region [127,128]. In the second inguinal–scrotal phase, the testis descends into the scrotum through the inguinal canal, a process dependent on testosterone that also appears to involve the INSL3/RXFP2 signal [129]. In INSL3 gene knockout mice, the ligaments did not develop, and the testes seemed to move loosely within the abdominal cavity, which leads to cryptorchidism [129].

Additionally, INSL3 has been found to inhibit germ cell apoptosis by binding to leucine-rich repeat-containing G protein-coupled receptor 8 (LGR8), which is expressed in germ cells [130,131,132]. Furthermore, studies have shown that higher serum INSL3 concentrations in men treated with hormonal male contraceptive regimens are associated with persistent sperm production [131]. These findings collectively indicate a strong association between INSL3 and spermatogenesis.

### 3.7. Estrogen

In recent years, research has uncovered the presence of estrogen, a hormone traditionally associated with females, in male reproductive processes. One of the key enzymes involved in estrogen production, known as CYP19A1 or aromatase, plays a crucial role in the conversion of androstenedione and testosterone into estrone and estradiol. This enzyme has been found to be expressed in Leydig cells, germ cells, and epididymal sperm [133]. When it comes to male reproduction, estrogen has direct effects on the development of male reproductive organs and the process of spermatogenesis. Studies on males with estrogen receptor-α (ERα) gene knockout have shown that the efferent ductule epithelium, which connects the testis to the initial segment of the epididymis, fails to absorb fluid properly [134,135]. This leads to backpressure atrophy in the testes and damages the process of spermatogenesis. Similar effects have also been observed in pubertal male mice that were treated with anti-estrogen compounds [136,137]. On the other hand, mice lacking the aromatase enzyme experience post-meiotic defects around 18 weeks of age, including increased apoptosis and reduced fertility, which can be improved by supplementing their diet with phytoestrogens [138,139,140].

However, adult male rats exposed to a diet high in phytoestrogens also experienced increased germ cell apoptosis and disruptions in spermatogenesis [141]. The involvement of estrogen in the development of prostatic hyperplasia has also been demonstrated [142]. Neonatal exposure to high levels of estrogen may permanently alter prostate development and differentiation [133]. This effect may be attributed to the potential reprogramming of prostatic stem and progenitor cells as a result of early estrogen exposure, leading to changes in their proliferation status [143]. Despite the recognized detrimental impact of estrogen on male fertility over the years, the appropriate expression of estrogen still plays a significant role in male reproductive capability and should not be overlooked.

### 3.8. Prolactin (PRL)

PRL is a peptide hormone secreted by lactotroph cells of the anterior pituitary gland, which plays a crucial role in female reproductive physiology. The release of PRL is regulated by various factors within the HPG axis, and it, in turn, regulates gonadal function by regulating the number of LH receptors on testicular Leydig cells and the release of pituitary gonadotropins [144]. In immature male rats, the prolonged suppression of PRL can inhibit the process of spermatocyte–spermatid conversion, alter Leydig cell morphology, and increase serum LH levels. These effects can be mitigated by administering exogenous PRL [145]. Furthermore, mice lacking the PRL gene showed no impaired fertility, but there were reductions in dopamine, LH, and FSH levels [146]. Conversely, male hyperprolactinemia (HPRL), often resulting from a pituitary tumor, is associated with erectile dysfunction (ED), decreased libido, and dysfunction in orgasm or ejaculatory ability [147,148]. Although studies have shown the effects of impaired PRL function on male reproduction, sexuality, and metabolism, the role of prolactin and its receptors in males is still unclear [144,149].

### 3.9. Oxytocin (OT)

OT is a neurohormone stored and secreted by the posterior pituitary gland and has been found to be present in the male reproductive tract [150]. Research has shown that OT promotes spermiation and sperm transfer, potentially by regulating steroidogenesis and the contractility of the seminiferous tubule [151,152]. Additionally, OT has been found to stimulate the secretion of testosterone and GnRH [153]. Interestingly, treatment with OT has been found to increase the expression of proliferating cell nuclear antigen (PCNA) and B cell lymphoma 2 (Bcl-2) proteins in the testis, suggesting that OT may play an important role in the proliferation, survival, and death of germ cells [152]. Although the molecular mechanism of the effect of OT on testicular activity is not clear, further exploration of the function and mechanism of OT in the testis is also of significant importance.

## 4. The HPG Axis in Hormonal Regulation

The HPG axis is the center of human reproduction, and its activation depends on the secretion of GnRH by the hypothalamus follows a pulsatile pattern. This pulsatile GnRH release was recently found to be caused by a group of kisspeptin neurons in the arcuate nucleus (ARC) of the hypothalamus [154]. The frequency and amplitude modulation of these pulses control the release of LH and FSH, which triggers key events in the reproductive system. The HPG axis is active during critical stages of life, including fetal development (midgestational fetus), infancy (minipuberty), and puberty. At each stage, the activation of the HPG axis serves different biological functions related to growth, sexual maturation, and reproductive capability (Table 2).

### 4.1. HPG Axis in Fetal Life

During embryonic development, specifically around 42 days of gestation, neurons that secrete GnRH migrate from the epithelium of the medial olfactory pit. They travel along nerve fibers that are rich in neural cell adhesion molecule (N-CAM) to reach the fetal hypothalamus [159]. GnRH expression begins in the first trimester, while levels of LH and FSH can be detected in the fetal serum and pituitary gland, starting at approximately 12 weeks of gestation in the second trimester (Figure 4). Towards the end of gestation, both LH and FSH levels decrease and become very low at term [155,160]. This decrease may be attributed to the inhibitory effects of high levels of placental estrogen on HPG axis activity at the end of pregnancy [161]. At the eighth week of pregnancy, fetal testes begin to secrete T, and by 10–20 weeks, testosterone levels reach those typically found in adults. However, testosterone levels decline towards the end of pregnancy. During this stage, the testosterone stimulation of the sensory branch of the genitofemoral nerve causes rhythmic contractions of the gubernaculum that pull the testicles into the scrotum as the first process of testosterone regulation during development [162]. Additionally, AMH is produced in the eighth week of pregnancy and causes the regression of the Müllerian ducts, which prevents the formation of the uterus and fallopian tubes [163]. The attainment of proper masculinization after birth hinges on the adequate development of testicular somatic and germ cells during the fetal period, particularly during a pivotal phase termed the masculinization programming window. This critical developmental interval sets the stage for future reproductive health in adult males, and disruptions during this period can have lasting consequences. Consequently, many reproductive health issues encountered in adulthood may trace their roots back to fetal development [69,164]. Understanding the intricate processes and regulatory mechanisms at play during this window is essential for the prevention and management of male reproductive disorders.

### 4.2. HPG Axis in Minipuberty

In boys, the HPG axis becomes active again after birth. Correspondingly, other hormones also undergo changes throughout life under the influence of the HPG axis. Gonadotropin levels remain high for the first three months of life but start to decrease by the sixth month. During this time, testosterone levels reach their peak at one to three months of age [165,166]. This stage is referred to as minipuberty, and it plays a crucial role in the development of the fetus’s penis and testicles. After minipuberty, the HPG axis becomes inactive until puberty [167]. During minipuberty, gonadotropins induced by GnRH stimulate Leydig cells to actively release androgen, while Sertoli cells secrete high levels of AMH. A study using a stereological approach suggests that the increase in Sertoli cell numbers primarily occurs during this period and puberty [168]. At this time, germ cells go through mitosis but do not enter meiosis [169]. Although Sertoli cells and germ cells remain immature before the onset of puberty, the size and potential for sperm production in adult rat testes are largely determined by the proliferation of testicular cells during fetal and neonatal development [170].

### 4.3. HPG Axis in Puberty

In mammals, the testicles contain Sertoli cells and undifferentiated spermatogonia from infancy until puberty, when spermatogonia begin to differentiate and undergo meiosis. Before the onset of puberty, the HPG axis remains relatively dormant during childhood, with GnRH, LH, and FSH being released in a pulsatile and nocturnal pattern and maintained at very low levels [171,172]. During puberty, the signal transduction of kisspeptin and neurokinin B (NKB) to GnRH neurons is further increased, resulting in gonadotropin secretion and gonadal maturation [171,173]. At this point, LH stimulates the differentiation of Leydig cells and produces testosterone. The combined effect of testosterone and FSH stimulation on Sertoli cells eventually leads to the initiation of spermatogenesis [102]. At the same time, there is a significant increase in the volume of the testis and the diameter of seminiferous tubules due to germ cell proliferation, which is often the first indicator of puberty [174]. Testosterone promotes the maturation of Sertoli cells and also reduces AMH production. The decrease in AMH levels is closely linked to an increase in inhibin B during early puberty [175]. In the early stage of testicular maturation, the concentration of inhibin B is positively correlated with the level of FSH, while in the middle to late stage of adolescence, it shows an inverse relationship with FSH and slightly decreases [176,177] (Figure 4). However, the concentration of inhibin B is significantly correlated with the entire adolescent testicular volume [176], while the serum concentration of INSL3 gradually increases throughout puberty, and this increase is dependent on LH [178]. This intricate hormonal interplay is pivotal in orchestrating the developmental changes in the testes during puberty.

### 4.4. HPG Axis in Adulthood

Male fertility peaks at puberty, when the reproductive system is fully developed, but the regulation of the HPG axis remains important in adulthood. In adulthood, the normal androgen levels maintained by the HPG axis are not only the basis for spermatogenesis, sexual function, and libido but also crucial for maintaining muscle mass, bone health, and erythropoiesis [179,180,181]. Consequently, disruptions in the HPG axis can have implications beyond reproductive health, potentially affecting the skeletal and cardiovascular systems as well [180,181].

## 5. Hormone Disorders and Male Hormonal Contraceptive

### 5.1. Hypogonadism

Male infertility is characterized by abnormal sperm parameters and can be divided into three categories: testicular dysfunction (possibly associated with primary hypogonadism), hypothalamic–pituitary diseases (resulting in secondary hypogonadism), and the obstruction of semen outflow (obstructive azoospermia, OA) [182]. About 40% of infertile couples are affected by male hypogonadism [183].

Male hypogonadism is generally defined as low circulating testosterone levels, with signs and symptoms of testosterone deficiency. Serum testosterone levels are the primary indicator for detecting hypogonadism in adult males. Generally, repeated measurements of total testosterone concentration in the morning serum < 9.7–10.4 nmol/L and free testosterone < 0.17–0.31 nmol/L can be used as a reference value [184,185]. With decreased serum concentrations of total testosterone or free testosterone level, elevated LH and FSH levels should raise suspicion for a diagnosis of hypergonadotropic hypogonadism (primary hypogonadism), and low or inappropriately normal LH and FSH levels suggest hypogonadotropic hypogonadism (secondary hypogonadism) [186,187].

Primary hypogonadism is a failure of testicular testosterone production manifested by a significant decrease in testosterone concentrations and an increase in gonadotropin concentrations. Generally, primary hypogonadism can be caused by certain hereditary disorders (Klinefelter syndrome and other rare chromosomal abnormalities), infections, testicular injuries (torsion and trauma), orchitis, drugs, or congenital anorchidism [188]. Secondary hypogonadotropin refers to the failure of gonadotropin secretion in the central nervous system, mainly caused by hypothalamic and pituitary disorders (including prolactinoma and other pituitary tumors), resulting in decreased spermatogenesis, low serum testosterone levels, and low or inappropriate normal LH and FSH levels [185,187]. Hemochromatosis, morbid obesity, hyperprolactinemia, medications (i.e., glucocorticoids and opioids), traumas, eating disorders, anabolic steroid abuse, or other genetic disorders (Prader–Willi syndrome and Kallmann’s syndrome) are also factors leading to secondary hypogonadism [188,189].

### 5.2. Male Hormonal Contraceptive

Although there are many methods of contraception, the unintended pregnancy rate in the United States remains at about 45%, and many people are reluctant to use long-acting reversible contraceptives (the IUD and the implant) despite their failure rate of only 1% [190,191]. There are not many reversible contraceptive methods available to men, including the male condom and withdrawal, with failure rates of 13% and 20%, respectively [191]. Developing new contraceptives for men is challenging, and interfering with spermatogenesis through hormonal approaches is a hot topic of research. It is well known that intratesticular concentrations of testosterone (ITT) are 50 to 100 times higher than those in the peripheral circulation [192], which is thought to be essential for spermatogenesis and has been proposed as a new way of developing hormonal male contraceptives [193,194]. The use of exogenous steroids alone or in combination with progestins or GnRH agonists or antagonists can inhibit testicular testosterone production by the feedback inhibition of the HPG axis while also maintaining adequate serum levels to ensure other androgen-related functions [195,196].

However, oral testosterone is cleared too rapidly, hampering the search for safe and effective oral androgens [195]. Therefore, there is a need for the development of new male hormonal contraceptives that are convenient, effective, reversible, and affordable. Testosterone undecanoate (TU), a testosterone ester created by the fatty acid esterification of testosterone, has been approved for use in several countries. It has been proven in clinical trials to be a highly reversible, effective, and safe male contraceptive, with a total efficacy was about 95% [197]. 7α-methyl-19- nortestosterone (MENT, an androgenic–anabolic steroid) has been proven to be more effective in inhibiting pituitary gonadotropin than testosterone and the potential of its subcutaneous implants is being investigated [198,199]. Dimethandrostenolone undecanoate (DMAU) and 11β-methyl-19-norT (11β-MNT) are structurally similar and are currently under clinical investigation [200]. Both of them bind avidly to AR and have proved to be safe and well tolerated without serious side effects and reversibly inhibit the hypothalamic–pituitary–testicular axis [201,202]. In addition, randomized, double-blind clinical trials demonstrated that a combination of testosterone and nestorone (NES, a nonandrogenic progestin) transdermal gel inhibited spermatogenesis, with 88.5% of men using the NES + T gel daily inhibiting sperm concentration to 1 million/mL or lower with minimal adverse effects [203].

Hormonal male contraception has been shown to be effective in clinical trials, but a novel male contraceptive will mitigate a myriad of biopsychosocial risks by male users and their partners [204]. There must be further testing in long-term studies to determine whether these male hormones contraceptives are safe and effective in inhibiting sperm production. Realistically, long-term trials on a sufficient number of couples can take years before the product can be brought to market.

### 5.3. Research and Future Perspective

Although research began decades ago, our understanding of male hormone regulation is continually evolving with new discoveries. A recent study revealed that in LH receptor knockout mice, the resumption of spermatogenesis and sexual function required only approximately 5 nmol/L of circulating testosterone and intratesticular testosterone (ITT) concentration, suggesting that elevated ITT levels may be an inherent characteristic of the testis as a site of testosterone production [205]. This finding challenges the traditional view of hormone dynamics in male fertility. Additionally, hormones play a crucial role in sexual function and spermatogenesis, and while there has been extensive research on the targets and functions of hormonal regulation, the endeavor to simulate these processes in vitro presents significant challenges. The production of sperm in vitro is a complex process that requires the accurate simulation of the physiological and hormonal signals within the natural environment of spermatogonial stem cells (SSCs). These cells are essential for spermatogenesis and necessitate a precise combination of hormones, growth factors, and other signals for successful development [206]. However, the complexity of both endogenous and exogenous factors that influence aging and related diseases [207], as well as the challenge of developing high-throughput assays to evaluate the impacts of endocrine-disrupting chemicals [208], makes the task of mimicking these signals in vitro particularly daunting. Therefore, engineering reproductive tissues outside the body faces challenges in replicating hormonal fluctuations and re-establishing the intricate physiological interactions found in natural settings [209]. This complexity is not unique to reproductive biology; for instance, the in vivo intestinal stem cell compartment also illustrates the challenges in capturing physiological interactions within tissues [210]. Nevertheless, the endocrinology of the male reproductive system is particularly critical as it regulates the synthesis of sperm and other male reproductive entities [211]. The regulation is intertwined with the endocrine hormones produced within the male reproductive system, highlighting the close relationship between hormonal action and spermatogenesis [211].

Approaches such as testicular tissue or SSC transplantation, and in vitro spermatogenesis, are emerging as vital strategies to restore fertility in individuals suffering from conditions like cancer or azoospermia [212]. Promising results have been shown in mice, where neonatal fresh and cryopreserved mouse testicular tissue fragments were able to undergo spermatogenesis and produce viable spermatozoa [213,214]. However, achieving similar success in primates has remained elusive. One of the core difficulties lies in our limited understanding of the germ cell niche—its structural and nutritional requirements—and the complex regulatory mechanisms governing spermatogenesis. Hormonal regulation within this process is notably challenging to dissect due to the interconnected nature of the endocrine system, where hormones often interact with or influence other testis-acting hormones in largely synergistic ways.

In conclusion, in vitro sperm production is a highly intricate venture, contingent upon the replication of precise physiological and hormonal signals inherent to the natural developmental niche. Acknowledging and addressing the challenges in reproducing these signals is critical for advancing our capabilities in fertility restoration and for deepening our understanding of the elaborate interplay of physiological systems involved in sperm production. Future research should employ high-resolution, multidimensional detection and analytical methods to delve into the downstream regulatory networks and molecular underpinnings of male hormone regulation. Such studies could elucidate the dynamic effects of hormones during development, potentially leading to significant advancements in the diagnosis, classification, and treatment of male infertility. This could also pave the way for the discovery of a broader range of effective therapeutic strategies and medications.

## 6. Conclusions

The male reproductive system’s development and function are intricately linked to complex hormonal interactions, with the HPG axis’s pulsatile hormone release playing a key role in male fertility. Early diagnosis and intervention are vital for addressing hormonal disorders that can impact reproductive health and well-being. Understanding the complex regulatory networks and molecular mechanisms in testicular development and spermatogenesis is key to creating innovative treatments and contraceptives. Ongoing research is vital for advancing contraceptive options and enhancing overall reproductive health.

## Figures and Tables

**Figure 1 ijms-25-05805-f001:**
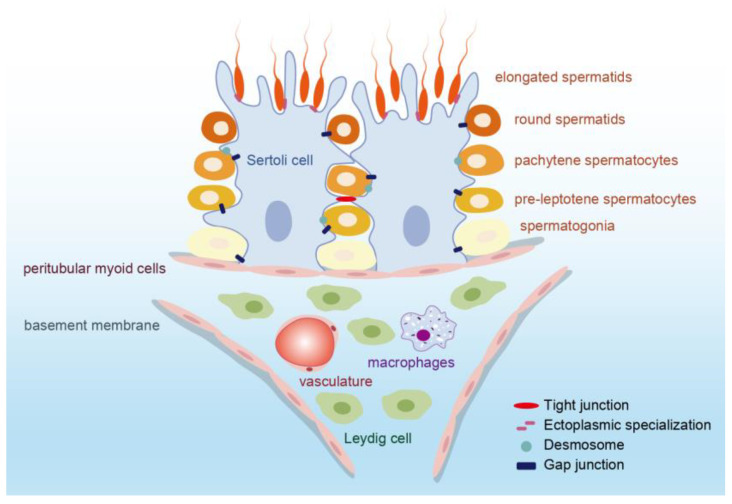
Schematic representation of the structure of the seminiferous tubules. There are somatic Sertoli cells and germ cells in the seminiferous tubules and Leydig cells, blood vessels, and immune cells in the interstitium. SSCs and Sertoli cells are attached to the basement membrane. In the process of spermatogenesis, germ cells (spermatogonia, pre-leptotene spermatocytes, pachytene spermatocytes, round spermatid, and elongating spermatid) move in the seminiferous epithelium and undergo meiosis until they are released to the seminiferous tubular fluid. There are four types of cell junctions in the seminiferous epithelium. The junctions between Sertoli cells form the BTB, which provides an immunologically privileged environment for spermatogenesis.

**Figure 2 ijms-25-05805-f002:**
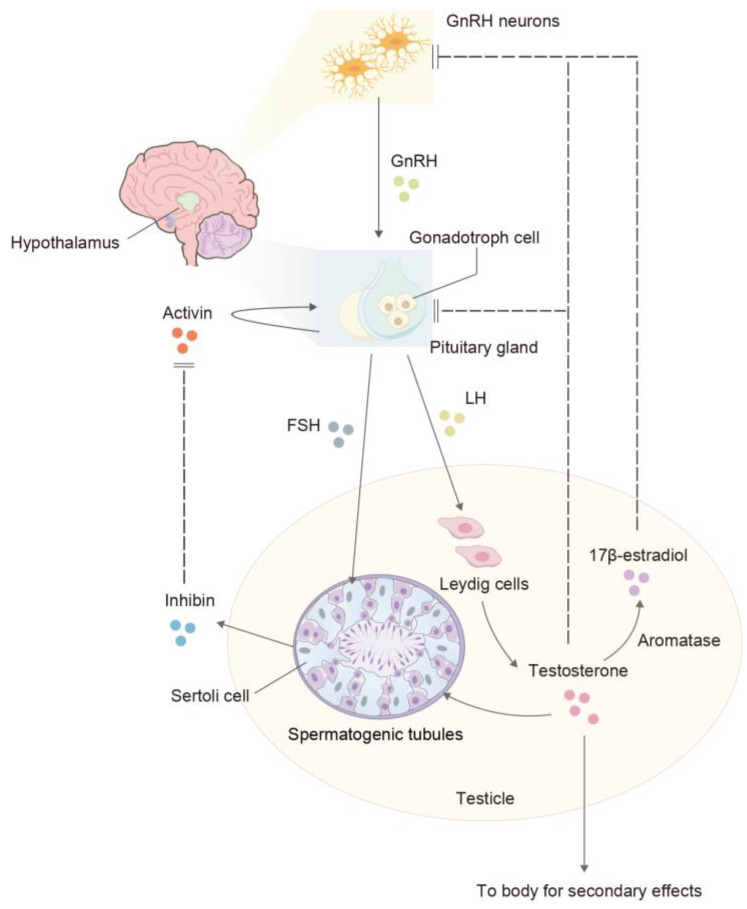
The HPG axis. The hypothalamus regulates the biosynthesis and secretion of pituitary hormones LH and FSH through GnRH. LH induces Leydig cells to secrete testosterone, which reduces GnRH and LH production through negative feedback. FSH acts on Sertoli cells to induce the secretion of inhibin, which in turn inhibits the production of FSH. Sertoli cells, under the combined action of testosterone and FSH, stimulate the proliferation and maturation of germ cells. Testosterone is also essential for promoting muscle and bone development. Furthermore, testosterone is metabolized by aromatase (CYP19A1) into 17β-estradiol (E2) which exerts inhibition on the hypothalamus. Normal spermatogenesis depends on the coordinated action of all these hormones.

**Figure 3 ijms-25-05805-f003:**
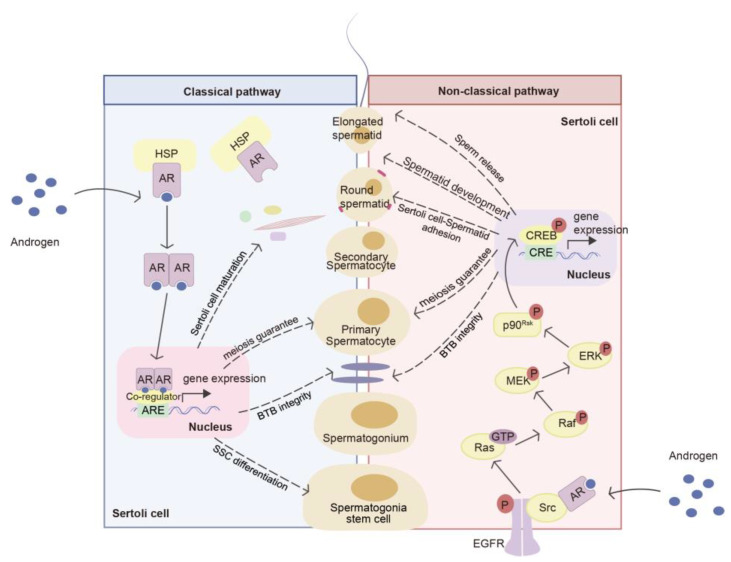
The classical and non-classical pathways of androgen between the two neighboring Sertoli cells allow the germ cells to continuously mature and move toward the lumen. The classic androgen pathway is shown in the Sertoli cells on the left. Androgen diffuses into the cytoplasm through the plasma membrane and interacts with HSP-binding AR, then the AR dissociates from HSP and localizes to the nucleus. In the nucleus, AR homodimers recruit co-regulators and bind to the ARE of the target gene to regulate their transcription. The classic pathways guarantee Sertoli cell maturation, BTB integrity, SSC differentiation, and spermatocyte meiosis. The non-classical androgen pathway is depicted within the Sertoli cells on the right. Androgen binds to AR on the cell membrane and subsequently interacts with Src. Src then activates EGFR, initiating a series of cascade reactions. Ultimately, the phosphorylated p90Rsk translocates to the nucleus and activates CREB to bind to the CRE of the target genes, thereby regulating their transcription. This process enables androgens to modulate the BTB integrity of Sertoli cells, the meiosis of spermatocytes, Sertoli cell–spermatid adhesion, spermatid development, and sperm release.

**Figure 4 ijms-25-05805-f004:**
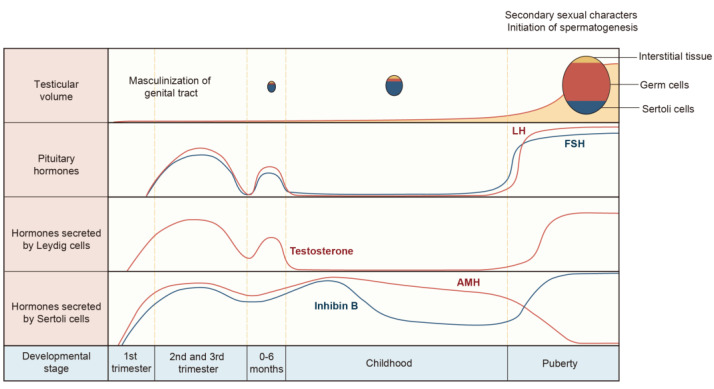
Developmental physiology of the testis and serum levels of gonadotropins, testosterone, inhibin B, and AMH from the fetal stage to puberty. The activation of the male HPG axis occurs in three distinct waves. The first wave commences during the fetal period, where testosterone expression prompts the differentiation of internal genitalia and the masculinization of the external genitalia. Additionally, AMH facilitates the regression of Müllerian ducts. This process reaches its peak in mid-pregnancy and decreases at birth. Following birth, there is a brief reactivation of the HPG axis during minipuberty. This second wave is characterized by an increased secretion of LH and FSH, leading to a testosterone peak comparable to that of puberty. Subsequently, during infancy and childhood, levels of gonadotropins and testosterone decrease, whereas levels of AMH and inhibin B continue to rise. Testicular volume experiences a slight increase during fetal and childhood stages, primarily due to the proliferation of Sertoli cells. It is not until puberty that the HPG axis is reactivated. During puberty, testosterone stimulates Sertoli cell maturation and initiates spermatogenesis, accompanied by a significant increase in testicular volume. Concurrently, testosterone inhibits AMH levels, and FSH stimulation elevates inhibin B levels.

**Table 1 ijms-25-05805-t001:** Hormone secretory regions and target cells.

Hormones	Secretory Regions	Target Cells	Disordered Diseases
GnRH	hypothalamic neurosecretory cells	pituitary gonadotrophs	precocious puberty, hypergonadism, Kallmann syndrome, oligospermia [20]
LH	pituitary gonadotrophs	Leydig cells	hypogonadism [21]
FSH	pituitary gonadotrophs	Sertoli cells, peritubular myoid cells, Spermatogoniums	hypogonadism [21]
T	Leydig cells, Sertoli cells	germ cells, Sertoli cells, PMCs	TDS, micropenis [21,22]
AMH	Sertoli cells	Müllerian ducts mesenchymal cells	persistent Müllerian duct syndrome (PMDS) [23]
Inhibin B	Sertoli cells	pituitary gonadotrophs	spermatogenesis disorder [24]
INSL3	Leydig cells	Leydig cells	cryptorchidism [25]

**Table 2 ijms-25-05805-t002:** The characteristics and function of hormones in different life stages.

Hormones	Fetal Life	Minipuberty	Puberty
GnRH	serum LH and FSH levels in the second trimester are independent of GnRH, and then GnRH gradually controls the release of LH and FSH [155]	stimulates Sertoli cells to secrete inhibin B and AMH, and Leydig cells to produce INSL3	increases gradually, triggering the secretion of LH and FSH
LH	replaces HCG to promote the secretion of testosterone by Leydig cells [21]	stimulates Leydig cells to release testosterone	stimulated the differentiation of Leydig cells and their ability to produce testosterone
FSH	stimulates Sertoli cell proliferation and increases AMH and inhibin B	stimulates Sertoli cell proliferation and increases AMH and inhibin B	stimulates the proliferation of immature Sertoli cells and spermatogonia
T	induces the differentiation and development of the mesonephric duct into seminal vesicles, epididymis, and spermaduct [156]	promotes the conversion of germ cells into spermatogonia	initiation of spermatogenesis
AMH	causes fallopian tube regression in men, preventing the formation of the uterus and fallopian tubes [157]	as a diagnostic indicator of male fertility-related disorders [158]	as a diagnostic indicator of male fertility-related disorders [158]
Inhibin B	regulates FSH secretion and acts as a marker for Sertoli cell function	regulates FSH secretion and acts as a marker for Sertoli cell function	inhibit FSH secretion and markers of sperm production in men
INSL3	the regulation of intra-abdominal testicular descent by regulating the growth and differentiation of the gubernaculum [129]	as an accurate measure of Leydig cell functional capacity [125]	as an accurate measure of Leydig cell functional capacity [125]

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
