# Peer review of "Hormone Regulation in Testicular Development and Function"

_ijms, 2024, doi:10.3390/ijms25115805_

Round 1
Reviewer 1 Report
Comments and Suggestions for Authors
The manuscript details testicular physiology and hormonal regulation. I don't have many comments about the text in general. However, I suggest to the authors, in item "5.3 Research and future perspective", to add some research with in vitro cells, which demonstrates the attempt to produce sperm in vitro and also the difficulties and complexity of producing an in vitro niche that contains all the physiology/hormonal.
Comments on the Quality of English LanguageNo comments
Author Response
Thank you for your insightful comments and constructive feedback on our manuscript. We appreciate the opportunity to discuss the attempt to produce sperm in vitro and also the difficulties and complexity of producing an in vitro niche that contains all the physiology/hormonal.
In the revised manuscript, we provided a discussion on in vitro spermatogenesis and the direction of future research. Please check lines 630-674.
Lines 630-674:
Additionally, hormones play a crucial role in sexual function and spermatogenesis, and while there has been extensive research on the targets and functions of hormonal regulation, the endeavor to simulate these processes in vitro presents significant challenges. The production of sperm in vitro is a complex process that requires the accurate simulation of the physiological and hormonal signals within the natural environment of spermatogonial stem cells (SSCs). These cells are essential for spermatogenesis and necessitate a precise combination of hormones, growth factors, and other signals for successful development [1]. However, the complexity of both endogenous and exogenous factors that influence aging and related diseases [2], as well as the challenge of developing high-throughput assays to evaluate the impacts of endocrine-disrupting chemicals [3], make the task of mimicking these signals in vitro particularly daunting. Therefore, engineering reproductive tissues outside the body faces challenges in replicating hormonal fluctuations and re-establishing the intricate physiological interactions found in natural settings [4]. This complexity is not unique to reproductive biology; for instance, the in vivo intestinal stem cell compartment also illustrates the challenges in capturing physiological interactions within tissues[5]. Nevertheless, the endocrinology of the male reproductive system is particularly critical as it regulates the synthesis of sperm and other male reproductive entities [6]. The regulation is intertwined with the endocrine hormones produced within the male reproductive system, highlighting the close relationship between hormonal action and spermatogenesis [6].
Approaches such as testicular tissue or SSC transplantation, and in vitro spermatogenesis, are emerging as vital strategies to restore fertility in individuals suffering from conditions like cancer or azoospermia [7]. Promising results have been shown in mice, where neonatal fresh and cryopreserved mouse testicular tissue fragments were able to undergo spermatogenesis and produce viable spermatozoa [8, 9]. However, achieving similar success in primates has remained elusive. One of the core difficulties lies in our limited understanding of the germ cell niche—its structural and nutritional requirements—and the complex regulatory mechanisms governing spermatogenesis. Hormonal regulation within this process is notably challenging to dissect due to the interconnected nature of the endocrine system, where hormones often interact with or influence other testis-acting hormones in largely synergistic ways.
In conclusion, in vitro sperm production is a highly intricate venture, contingent upon the replication of precise physiological and hormonal signals inherent to the natural developmental niche. Acknowledging and addressing the challenges in reproducing these signals are critical for advancing our capabilities in fertility restoration and for deepening our understanding of the elaborate interplay of physiological systems involved in sperm production. Future research should employ high-resolution, multidimensional detection and analytical methods to delve into the downstream regulatory networks and molecular underpinnings of male hormone regulation. Such studies could elucidate the dynamic effects of hormones during development, potentially leading to significant advancements in the diagnosis, classification, and treatment of male infertility. This could also pave the way for the discovery of a broader range of effective therapeutic strategies and medications.
References:
- Ibtisham, F.; Honaramooz, A., Spermatogonial Stem Cells for In Vitro Spermatogenesis and In Vivo Restoration of Fertility. Cells 2020, 9 (3).
- Guo, J.; Huang, X.; Dou, L.; Yan, M.; Shen, T.; Tang, W.; Li, J., Aging and aging-related diseases: from molecular mechanisms to interventions and treatments. Signal Transduct Target Ther 2022, 7 (1), 391.
- Diamanti-Kandarakis, E.; Bourguignon, J.-P.; Giudice, L. C.; Hauser, R.; Prins, G. S.; Soto, A. M.; Zoeller, R. T.; Gore, A. C., Endocrine-disrupting chemicals: an Endocrine Society scientific statement. Endocr Rev 2009, 30 (4), 293-342.
- Gargus, E. S.; Rogers, H. B.; McKinnon, K. E.; Edmonds, M. E.; Woodruff, T. K., Engineered reproductive tissues. Nat Biomed Eng 2020, 4 (4), 381-393.
- Malijauskaite, S.; Connolly, S.; Newport, D.; McGourty, K., Gradients in the in vivo intestinal stem cell compartment and their in vitro recapitulation in mimetic platforms. Cytokine Growth Factor Rev 2021, 60, 76-88.
- O'Donnell, L.; Stanton, P.; de Kretser, D. M., Endocrinology of the Male Reproductive System and Spermatogenesis. In Endotext, Feingold, K. R.; Anawalt, B.; Blackman, M. R.; Boyce, A.; Chrousos, G.; Corpas, E.; de Herder, W. W.; Dhatariya, K.; Dungan, K.; Hofland, J.; Kalra, S.; Kaltsas, G.; Kapoor, N.; Koch, C.; Kopp, P.; Korbonits, M.; Kovacs, C. S.; Kuohung, W.; Laferrère, B.; Levy, M.; McGee, E. A.; McLachlan, R.; New, M.; Purnell, J.; Sahay, R.; Shah, A. S.; Singer, F.; Sperling, M. A.; Stratakis, C. A.; Trence, D. L.; Wilson, D. P., Eds. MDText.com, Inc.
- Bashiri, Z.; Gholipourmalekabadi, M.; Khadivi, F.; Salem, M.; Afzali, A.; Cham, T.-C.; Koruji, M., In vitro spermatogenesis in artificial testis: current knowledge and clinical implications for male infertility. Cell Tissue Res 2023, 394 (3), 393-421.
- Sato, T.; Katagiri, K.; Gohbara, A.; Inoue, K.; Ogonuki, N.; Ogura, A.; Kubota, Y.; Ogawa, T., In vitro production of functional sperm in cultured neonatal mouse testes. Nature 2011, 471 (7339), 504-507.
- Yokonishi, T.; Sato, T.; Komeya, M.; Katagiri, K.; Kubota, Y.; Nakabayashi, K.; Hata, K.; Inoue, K.; Ogonuki, N.; Ogura, A.; Ogawa, T., Offspring production with sperm grown in vitro from cryopreserved testis tissues. Nat Commun 2014, 5, 4320.
Reviewer 2 Report
Comments and Suggestions for Authors
Dear authors, thank you for providing such exhaustive work on testicular development, function, regulation, and diseases. However, the manuscript submitted does not have the structure of a scientific manuscript rather than of a book chapter. Indeed, the manuscript submitted lacks its own aim of the study and appears a comprehensive resume of everything that is known about testicular function.
As such, I suggest you find a different way to publish this work.
Author Response
Thank you for your constructive feedback on our manuscript. We recognize the importance of a well-defined structure and clear objectives in scientific writing. Your comments have prompted us to reevaluate our manuscript's presentation, and we have made the following revisions:
- We supplemented the purpose of this study. Please check lines 50-52.
Lines 50-52:
This review aims to provide an overview of the hormones involved in the regulation of male reproduction, as well as recent advancements in related clinical and applied research.
- We have adjusted the structure of the article and deleted redundant subsubtitles to make the article more concise. Please check lines 184, 217, 230, 255, 262, 279, 289, 299, 306.
- We provided a discussion on the difficulties and complexity of producing an in vitro niche that contains all the physiology/hormonal. Please check lines 630-674.
Round 2
Reviewer 2 Report
Comments and Suggestions for Authors
Thank you for providing a good revision.